# Cuticular and Exuvial Biomass and Nitrogen Economy During Assimilation and Growth of the American Grasshopper, *Schistocerca americana*

**DOI:** 10.3390/insects16030327

**Published:** 2025-03-20

**Authors:** Donald E. Mullins, Sandra E. Gabbert

**Affiliations:** Department of Entomology, Virginia Tech, Blacksburg, VA 24061, USA; sgabbert@vt.edu

**Keywords:** insect molting, exuvial biomass and nitrogen loss, insect growth

## Abstract

The American grasshopper is native to North America, with populations primarily occurring in the eastern United States and the Bahamas. Although it is not considered a severe agricultural pest, it is capable of causing significant damage to many kinds of crops and is considered to be the most destructive grasshopper in Florida. One aspect of this insect’s basic biology that has not been studied is the assimilation of nitrogen during their development from egg to adult. This report provides information on nitrogen assimilation during the growth process, along with an assessment of the amount of nitrogen lost during the molting processes. We found that the body weight for both males and females increased significantly during their development and that the nitrogen losses due to molting were also significant.

## 1. Introduction

The utilization of energy in a controlled, programmed manner constitutes the most basic requirement and property of living organisms [1], and the ecological and nutritional processes that are crucial to an animal’s growth, survival, and reproductive fitness have energetic costs [2]. Procuring and assimilating nitrogen for insect growth and reproductive processes may represent a significant component of an insect’s nutritional budget [3]. A major expenditure of nutritional nitrogen and carbon is invested in the production of chitin [4]. Insect cuticles are a group of extremely diverse materials, varying in thickness, strength, elasticity, and color [4,5,6]. The cuticular exoskeleton protects insects against abiotic and biotic factors [7]. A major reason for the evolutionary success of insects is their chitinous exoskeleton’s light weight, variable density, and functional elasticity that covers a range of more than eight orders of magnitude [6]. Over millions of years of evolution, the mechanical properties of insect cuticles have been adapted to accommodate a variety of structural requirements in keeping with their morphological, ecological, and physiological requirements [4,6,8,9,10].

The structure of the insect cuticle includes three basic layers—the envelope, the epicuticle, and the procuticle—and is composed mainly of chitin, a glucose amine, and proteins forming a variety of polymers that are physically interconnected [7,11,12]; the extent of this networking determines its physical properties [13]. Because the formation of arthropodan cuticles and their functions are quite complex, it can be viewed as a dynamic system, undergoing continual changes [9,11]. The conservation of cuticular materials is important to the nitrogen economy of arthropods. During the molting process, the conservation of nitrogenous materials is initiated during the pre-molt period when a new cuticle is being formed while the older cuticle is degraded and their components reabsorbed. In general, the less highly sclerotized (endocuticle) is digested and conserved, while the more highly sclerotized (exocuticle) is discarded at the end of the molt as the exuvium [9,12,14]. The exuvium itself may be consumed by some insects as a means for recycling its components as a nutrient resource. For example, Formosan termites immediately consume the shredded exuviae of their nest mates [15]. Mira [16] has reported that cockroaches “invariably” consume their exuviae during their larval life; the individuals maintained on low-protein diets showed the highest levels of exuvial consumption.

There have been a limited number of reports on the losses of nitrogen and biomass that occur in arthropods during molting and, among these reports, there have been very limited quantitative data on the nitrogen and biomass lost as exuviae. The objective of this study was to determine the amount of biomass and nitrogen lost by an insect as a consequence of molting. This included quantifying the amount of nitrogen contained in discarded exuviae and the total amount of biomass and nitrogen lost during the total assimilation and growth processes that occur during the maturation of both male and female *Schistocerca americana*. This species was used in this study because of its pest status and the variety of investigations into the genera to which it belongs.

## 2. Materials and Methods

### 2.1. Insect Rearing and Experimental Protocol

American grasshoppers *Schistocerca americana* (Drury) were obtained from the Virginia Tech Entomology greenhouse facilities and maintained at 30 ± 5 °C, ambient (uncontrolled) RH, 14/10 LD photoperiod, and reared on a diet of fresh Romaine lettuce supplemented with a grain meal containing antibiotics ad libitum [17]. Newly hatched first instars were obtained from the main rearing colony and maintained individually in modified plastic food containers (screened, 15 L × 11 W × 10 H cm) housed in an environmental chamber (30 ± 3 °C; 50 ± 5% RH; 14 L/10 D photoperiod). The immatures were fed Romaine lettuce, provided with H_2_O ad libitum, and checked 1–2 times daily for molting activity. When a molt occurred, the exuvium was removed and stored, and the event recorded. After the final molt, the adults were immediately frozen (−20 °C) for nitrogen analysis. Because of incubator space limitations and the time requirements associated with the daily maintenance of the caged individuals, two separate assays were conducted, each consisting of 35 individuals (totaling 70). If the newly produced exuvia appeared to be incomplete (parts missing), that individual and its exuvia were discarded and not included in the study. Based on this protocol, only 38 individuals qualified to be included in this study (22 males and 16 females). From those 38 individuals, 10 individuals of each sex were randomly selected for the biomass and nitrogen analyses.

### 2.2. Experimental Procedures

**Dry weights:** Early-to-late instar exuviae and whole-body adult samples were weighed using either an analytical balance (Mettler AE 163, Thermo Scientific, Waltham, MA, USA; mg range) or micro balance (Orion Cahn C-35, Thermo Scientific, Waltham, MA, USA; µg range) after drying the specimens with indicator desiccant beads (24 h at 84 °C). Prior to nitrogen analysis, the specimens were stored at −20 °C over silica gel beads, which had been activated at 100 °C.

**Nitrogen Determinations:** Whole-body early-to-late instars and adult samples were initially ground with dry ice in a glass mortar and pestle assembly, followed by grinding with dry ice for 30 s in a dental amalgamator to ensure homogeneity; subsamples (0.3–1.7 mg) from this mixture were used for analysis. Nitrogen was determined using a micro-Kjeldahl digestion procedure [18,19]. Confirmation tests of the quantitative reliability of the nitrogen analytical method were performed on sample spike-overs that yielded 95% recovery for 10, 20, and 30 µgN spike-overs of the samples ranging from 199 to 235 µg tissue dry weight.

### 2.3. Statistics

The mean populations of the stadia, biomass, and nitrogen content of the exuviae and adults of the *S. americana* were subjected to the following: (1) Kolmogorov–Smirnov test of normality; and (2) *t*-test for 2 independent means using a two-tailed test; HO: u1 − u2 = 0, α = 0.05.

## 3. Results

### 3.1. Morphological Size Changes and Development

The visual comparison of the *S. americana* body mass and exuvial mass losses during development are presented in Figure 1A,B, showing the relative size changes of males (A) and females (B) as they develop from egg to adult. The total development time from egg to adult was similar for both males (42.9 ± 0.7 days) and females (44.2 ± 0.6 days), and the individual instar stadia ranged from 6.6 to 11.1 days for males and from 6.2 to 12.7 days for females (Table 1).

### 3.2. Development Time

The stadia time intervals observed for both the males and females are shown in Table 1. There were no time differences between the male and female stadia. The total development time was 42.9 ± 0.7 days for the males and 44.2 ± 0.6 for the females.

### 3.3. Exuvial Biomass and Nitrogen Losses During Assimilation and Growth

It should be noted that the exuvial biomass measurements are inclusive of nitrogen, which are approximately 10% of the exuvial biomass. Figure 2A compares the exuvial biomass losses that occurred between each successive molt during the growth of the males and females. The total biomass losses from six molts was 36.0 mg for the males and 43.2 mg for the females. Figure 2B compares the nitrogen exuvial losses that occurred between each successive molt during the growth of the males and females. The total nitrogen loss from six molts was 3.7 mg for males and 4.6 mg for females.

### 3.4. Total Biomass and Nitrogen Assimilation and Growth Compared with Exuvial Biomass and Nitrogen Losses

Figure 3A compares the total biomass assimilation versus the exuvial losses during the growth of the males and females. The males lost 13% of their total assimilated biomass (281 mg/individual) during growth from the egg to the adult stage. Similarly, females also lost 12% of their total assimilated biomass (368 mg/individual) during their growth from the egg to the adult stage. Figure 3B compares the nitrogen assimilation versus the exuvial losses during the growth of males and females. Males lost 11% of their total assimilated nitrogen (38.3 mg/individual) during growth from the egg to the adult stage. Similarly, females lost 11% of their total nitrogen (44.7 mg/individual) during their growth from the egg to the adult stage.

## 4. Discussion

### 4.1. Energetics and Energy Budgets

All organisms must comply with an energy budget that is inclusive of their molecular, physiological, and ecological processes [1,2,3]. Simply put, if an organism cannot function within its energy budget, it will be unable to contribute to the next generation of its species. In 1981, Downer [1] provided a review of the energy budget components in insects as part of an international symposium on energy metabolism in insects. More recently, other researchers have contributed to discussions on energy budgets. Ledder [20] noted that dynamic energy models are the most ambitious of resource models in insects but claimed that a careful derivation of basic assumptions is needed to provide a new model that will yield a better perspective on energy budgets. Llandres et al. [21] reported on the dynamic energy budget for the whole life cycle of a holometabolous insect, which was a significantly challenging analysis because of the strikingly different life stages. Other published works have addressed broader animal energetics issues associated with ecological modeling [22,23]. Finally, Tomlinson et al. [2] noted that the field of ecological energetics was bringing comparative physiology out of the laboratory and making information more “assessable to field ecologists addressing real-world questions at many spatial and temporal scales”. They also noted that, in this era of unprecedented global environmental challenges, ecological energetics opens up the “tantalizing” prospect of a more predictive, mechanistic understanding of the drivers of the decline of threatened species, delivering process-based modeling approaches to natural resource management.

### 4.2. Conservation of Cuticular Nitrogen During Insect Molt Cycles

There are reports that provide some information on the investment/relative costs associated with arthropodan cuticles. Examples of these reports are as follows: (1) LaSalle et al. [24] observed that the mass of exuviae from spiders (*Lycosa watsoni*) ranged from 0.18 to 5.40 mg dry weight, representing 4.4 to 10.6% of their total weight; the exuvial mass, as a percentage of the body weight, increased in the amount of material lost during successive molts. (2) Lease and Wolf [25] measured the exoskeletal chitin mass of 245 insect species (15 orders and 91 families), reported that the exoskeletal chitin scaled isometrically with the dry body mass, and concluded that it comprised a constant portion of insect body mass regardless of the body size for most insect orders. (3) The researchers in [26] examined food energy expenditures in gypsy moths (*Lymantria dispar*) and found that the proportion of food energy expended in exuvial formation ranged from 0.5 to 2.5% of the caloric food intake values.

Estimating the contents of “discarded” exuviae is problematic since, in many cases, insects consume the exuviae soon after they are discarded. Therefore, some reports on the estimated mass/energy content of discarded exuviae in Orthoptera may be quite variable. Carefoot [27] provided a comprehensive analysis on the energy partitioning of all stages of the solitary generations of Desert Locusts (*Schistocerca gregaria*). He reported that the total combined male and female exuviae produced was 2.9% dry wt., representing 5.6 ± 0.2 kcal/gdwt (females) and 5.5 ± 0.3 kcal/gdwt (males). However, he acknowledged that the intact exuvia were difficult to retrieve due to consumption based on the analysis of the “uneaten moult bits”. Bailey and Reigert [28] reported that exuvial losses made up 6–11% of the secondary production in their study on the bioenergetics of a grassland grasshopper (*Encoptolophus sordidus costalis*), and Delvi and Pandian [29] reported that exuviae represented 4–5% of the assimilated food consumption in paddy field grasshoppers (*Oxya velox*).

The primary focus of our study was to evaluate both the total biomass (e.g., dry weight) and total nitrogen assimilation during the development of *S. americana*. Therefore, a concerted effort was made to use only individual replicates in which we observed no or minimal consumption of discarded exuviae. As anticipated, the biomass increases observed during development of the first instars to newly molted adults were quite significant [Figure 1A,B Legend: biomasses increased 7-fold for males and 9-fold for females]. The total body nitrogen increases correspond to those observed in the total biomass changes [Figure 1A,B Legend: nitrogen content increased by 23-fold (males) and 25-fold (females)].

The total biomass and nitrogen losses from the “discarded” exuviae observed during the same developmental time frame were quite significant (male: biomass = 13%, nitrogen = 10%; females: biomass = 12%, nitrogen = 10%, (Figure 3A,B). The average nitrogen content of the discarded exuviae for males and females was 10% (Figure 3A,B). We observed that, during culture rearing activities, a significant number of exuviae disappeared, which was particularly apparent when a cohort of individuals was somewhat synchronized. Based on the observations by the other research groups that have conducted similar studies to ours, exuvial consumption is a common phenomenon. An excellent example of this was reported by Mira [16], who observed that an analysis of the frass produced after feeding cockroaches on exuvial meals showed that over 58% of the nitrogen present in the exuviae was recycled, demonstrating that cockroaches digested some part of the exuviae and supporting the hypothesis that discarded recycled exuviae have a nutritional role. Rychtal et al. [30]. used studies of cockroaches as an example of the recycling of the organic waste materials produced by members of a cockroach colony. They presented a “game-theoretical model” that supported a “waste recycling hypothesis” where the organic waste matter produced by members of a group represented a valuable resource that was communally inherited and utilized by group members. The nutritional role that discarded exuviae may play most likely resides in the recycling of their nitrogen content facilitated by the microbial systems residing in an insect’s hindgut. The digestion of lignocellulosic materials similar in complexity to discarded exuviae may be facilitated by insect gut microbes [31,32]. Recent reports on some insect systems support the hypotheses that changes in their gut microbial systems may be flexible in adjusting the availability of the substrates that they make available in the digestive tract [32,33,34,35].

## 5. Conclusions

Measurements of the energetics associated with the assimilation and growth of developing *S. americana* indicate that the losses of biomass and nitrogen are quite significant during the maturation process, requiring a nutritional budget that can support these losses. The opportunity to recycle discarded exuvial nitrogen utilizing gut microbes might serve to compensate for some nitrogen losses.

## Figures and Tables

**Figure 1 insects-16-00327-f001:**
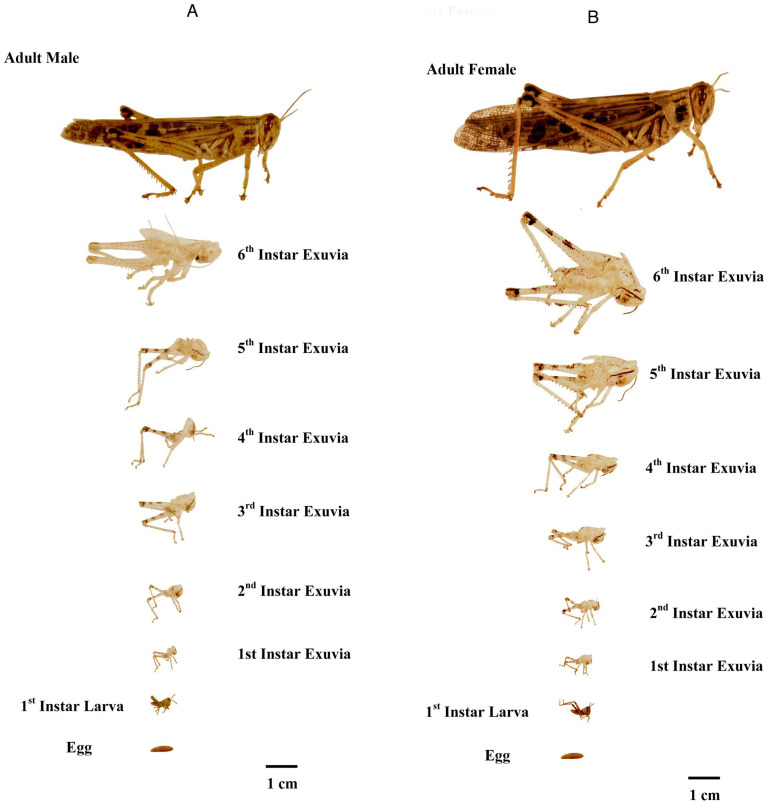
(**A**,**B**) Comparison of the size changes from egg to adult and exuvium size through development of the 6 instars. These images reflect the body mass and nitrogen increases that occurred during the assimilation and growth process. Biomasses increased by 7-fold for males and 9-fold for females. Nitrogen content increased by 23-fold for males and 25-fold for females.

**Figure 2 insects-16-00327-f002:**
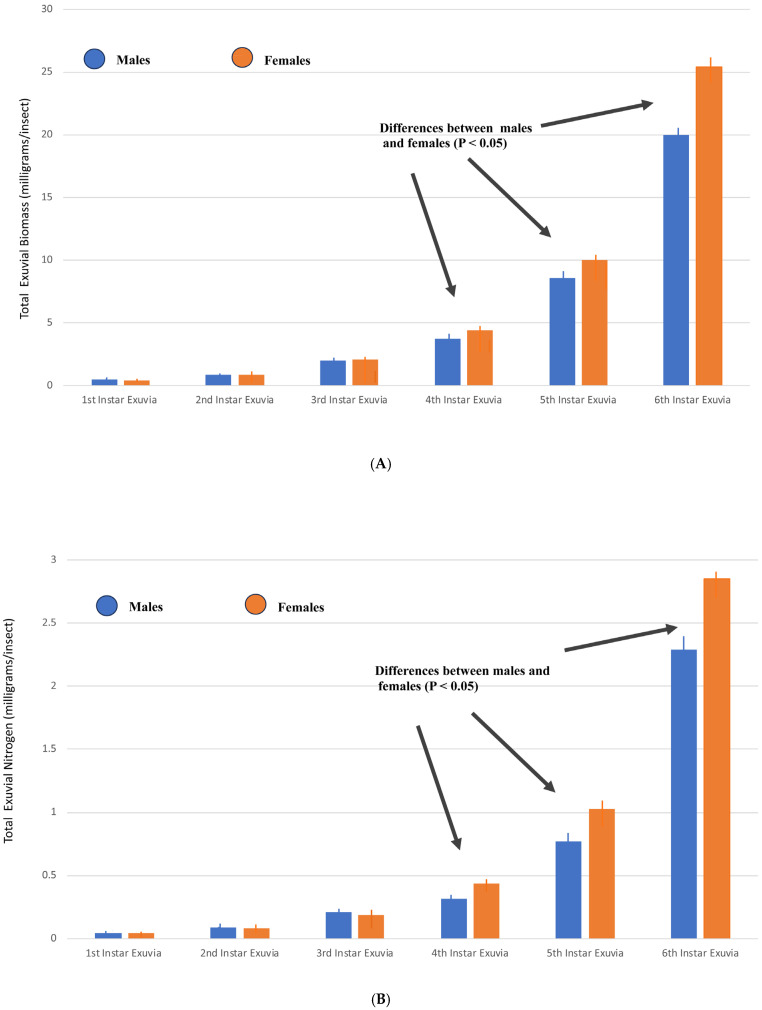
(**A**) Comparison of mean ± SEM exuvial biomass lost after each successive instar molt for both males and females (*n* = 10). These losses ranged from 0.45 ± 0.01 mg (1st instar male exuviae) to 20.0 ± 0.4 mg (6th instar male exuviae) for a total of 36 mg and from 0.40 ± 0.001 mg (1st instar female exuviae) to 25.4 ± 0.8 mg (6th instar female exuviae) for a total of 43.2 mg. The comparison of the exuvial biomasses between the males and females indicates there were differences in the last three instar exuvia (*t*-test for 2 independent means using a two-tailed test, *p* < 0.05). (**B**) Comparison of mean ± SEM exuvial nitrogen lost after each successive instar molt for both males and females (*n* = 10). These losses ranged from 0.046 ± 0.001 mg (1st instar male exuviae) to 2.29 ± 0.05 mg (6th instar male exuviae) for a total of 3.7 mg and from 0.042 ± 0.002 mg (1st instar female exuviae) to 2.85 ± 0.09 mg (6th instar female exuviae) for a total of 4.6 mg. The comparison of the exuvial nitrogen between the males and females indicates there were differences in the last three instar exuvia (*t*-test for 2 independent means using a two-tailed test, *p* < 0.05).

**Figure 3 insects-16-00327-f003:**
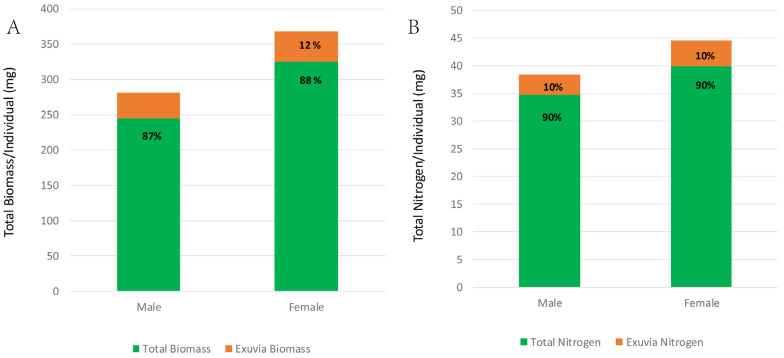
(**A**,**B**) Comparisons of the total assimilated biomass and nitrogen during male and female development with the amounts of their biomass and nitrogen discarded in their exuviae (*n* = 10). During the growth process, males lost 36.0 mg (12.6 ± 0.3%) and females lost 43.2 mg (11.8 ± 0.4%) of their total assimilated biomass as discarded exuviae (**A**). The amount of assimilated nitrogen lost from discarded exuviae for males was 3.7 mg (10.9 ± 0.5%) and for adult females the nitrogen lost was 4.6 mg (10.5 ± 0.5%) (**B**).

**Table 1 insects-16-00327-t001:** Comparison of time intervals between each stadia of male and female *Schistocerca americana* from egg hatch to newly molted adult.

Interval	*n*	^1^ Stadia Interval Average ± SEM Days	^1^ Stadia Interval Average ± SEM Days
		Males	Females
1st Instar	10	6.6 ± 0.2	6.2 ± 0.2
2nd Instar	10	5.2 ± 0.2	5.0 ± 0.2
3rd Instar	10	5.7 ± 0.2	5.7 ± 0.2
4th Instar	10	6.6 ± 0.2	6.4 ± 0.2
5th Instar	10	7.7 ± 0.3	8.2 ± 0.4
6th Instar	10	11.1 ± 0.3	12.7 ± 0.3

^1^ Statistical comparisons: Tests for differences between the male and female stadia were performed using a *t*-test for 2 independent means using a two-tailed test, *p* < 0.05.

## Data Availability

The raw data supporting the conclusions of this article will be made available by the authors on request.

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
