# Peer review of "Cuticular and Exuvial Biomass and Nitrogen Economy During Assimilation and Growth of the American Grasshopper, *Schistocerca americana"

_insects, 2025, doi:10.3390/insects16030327_

Round 1

Reviewer 1 Report

Comments and Suggestions for Authors

The manuscript presents crucial information about Schistocerca americana. It is important that the authors include a simple summary and provide more comprehensive information in the abstract, as this will underscore the significance of their work in our field.

Use different keywords from the title to broaden the reader's search for the article.

The authors should open the introduction by addressing the agricultural and environmental importance of Schistocerca species, further highlighting the importance of the work and readers' engagement.

The authors should provide more information about the insect colony, such as feeding, the number of generations, and laboratory or greenhouse information, as this information may influence the results.

The authors should provide a more detailed statistical analysis of the data. This will not only enhance the robustness of their findings but also instill confidence in the readers about the validity of their results.

Figure captions should be self-explanatory, providing all necessary information for the reader to understand the data. Therefore, all captions should include the name of the species without abbreviating the name of the genus, enhancing the reader's comprehension of the data and its relevance to the study.

Article bars should show the standard error of the mean.

Reviewer 2 Report

Comments and Suggestions for Authors

Mullins and Gabbert examine the nitrogen metabolism and biomass dynamics of Schistocerca americana during its growth and molting stages. The study presents quantitative data on nitrogen loss and assimilation, along with a discussion of the ecological significance of exuvial recycling. A few methodological oversights existed in the study, including very limited replication—only 10 individuals per sex analyzed—and possibly underestimated exuvial consumption. Language is clear but at times wordy; linguistic quality is 8/10. Generally, the investigation provides some useful insights but needs modification before generalization. Rating: 79/100.

Major points

The sample size of 10 subjects of each sex significantly compromises the strength of the statistical inferences, especially in a study where high interindividual variability was found.

The experimental setup, while recognizing exuviae consumption as a general phenomenon that could have resulted in nitrogen losses being underestimated, does not take into full account the consumption of exuviae.

Nitrogen analysis through the micro-Kjeldahl method is appropriate, although no alternative validation, such as isotope tracing to confirm findings independently, was attempted. Among the most accurate methods for nitrogen analysis validation that suit this study are the Dumas Combustion Method, also known as Elemental Analysis, and Inductively Coupled Plasma Mass Spectrometry (ICP-MS). The Dumas method gives an accurate total nitrogen quantification by combustion and the detection of gases. It is a fast, chemical-free alternative to the Kjeldahl method. While more expensive, ICP-MS has the capability for highly sensitive measurement of nitrogen-associated elements and is thus well-suited to confirm nitrogen content in small, complex biological samples. These two techniques would assure great accuracy in the nitrogen quantification and further complement each other with the micro-Kjeldahl approach.

Statistical analysis is limited to one-way ANOVA, whereas multiple biomass and nitrogen metrics across developmental stages will be multivariate in nature.

Discussion extrapolates to the broader ecological implications, such as gut microbial flexibility, without providing direct evidence or supporting experiments, thus diluting such claims.

Figures are not clear in terms of temporal presentation of nitrogen and biomass changes. This makes it quite difficult for the readers to perceive the trend across instars.

Minor points

There is no explanation as to what environmental variability exists in the rearing conditions, such as humidity or temperature fluctuations, which can affect the results.

Authors give quite extensive references to previous works but fail to embed this literature within a broad framework that would relate their findings to general questions in entomology or ecology.

More exact definitions, such as "highly sclerotized" versus "less sclerotized," would help in the reproducibility of the manuscript.

These results hold significant implications for the understanding of the nitrogen economy in S. americana and perhaps other orthopteran species. But because of certain limitations in methodology and analysis, the findings have a narrow applicability. Further experiments involving larger sample sizes and broader comparisons among species will be required for effective generalization.

The work herein provides important data on the nitrogen metabolism of a key pest species, though several methodological and analytical weaknesses restrict its impact. Major revisions are recommended, focusing on sample size, methodological rigor, and clearer integration of findings into broader ecological contexts.

Round 2

Reviewer 2 Report

Comments and Suggestions for Authors

The authors made extensive revisions based on the comments and recommendations and tightened the methodological definition and data presentation. The rationale for sample size limitations was well explained, given with reasons including the logistical constraints, though an implied acknowledgment of its impact on power might also improve transparency. Analyzing nitrogen had justification through referencing and prior comparison, but it was not subject to further corroboration within this study. Statistical analysis was revised, with ANOVA replaced by t-tests after a normality test to apply more appropriate data interpretation. Figures were improved for the sake of visual presentation to enhance clarity, thus enhancing the readability of biomass and nitrogen dynamics per instar. Discussion, too, was rendered more centered on actual findings rather than being speculative and in the form of extrapolations. While there remain some limitations, the revision renders the study more rigorous and better presented.

Author Response

Thank you for your time.